# The Impact of Withdrawn vs. Agitated Relatives during Resuscitation on Team Workload: A Single-Center Randomised Simulation-Based Study

**DOI:** 10.3390/jcm11113163

**Published:** 2022-06-02

**Authors:** Timur Sellmann, Andrea Oendorf, Dietmar Wetzchewald, Heidrun Schwager, Serge Christian Thal, Stephan Marsch

**Affiliations:** 1Department of Anaesthesiology and Intensive Care Medicine, Bethesda Hospital, 47053 Duisburg, Germany; t.sellmann@bethesda.de; 2Department of Anaesthesiology 1, Witten/Herdecke University, 58455 Witten, Germany; serge.thal@uni-wh.de; 3Institute of Emergency Medicine, 59755 Arnsberg, Germany; andrea.oendorf@rub.de (A.O.); d.wetzchewald@aim-arnsberg.de (D.W.); h.schwager@aim-arnsberg.de (H.S.); 4Department of Internal Medicine, Gertrudis Hospital, 45701 Herten, Germany; 5Department of Anaesthesiology, Helios University Hospital, 42283 Wuppertal, Germany; 6Department of Intensive Care, University Hospital, 4031 Basel, Switzerland

**Keywords:** cardiopulmonary resuscitation, team performance, randomised controlled trial, family presence, simulation, NASA task load index

## Abstract

Background: Guidelines recommend that relatives be present during cardiopulmonary resuscitation (CPR). This randomised trial investigated the effects of two different behaviour patterns of relatives on rescuers’ perceived stress and quality of CPR. Material and methods: Teams of three to four physicians were randomised to perform CPR in the presence of no relatives (control group), a withdrawn relative, or an agitated relative, played by actors according to a scripted role, and to three different models of leadership (randomly determined by the team or tutor or left open). The scenarios were video-recorded. Hands-on time was primary, and the secondary outcomes comprised compliance to CPR algorithms, perceived workload, and the influence of leadership. Results: 1229 physicians randomised to 366 teams took part. The presence of a relative did not affect hands-on time (91% [87–93] vs. 92% [88–94] for “withdrawn” and 92 [88–93] for “agitated” relatives; *p* = 0.15). The teams interacted significantly less with a “withdrawn” than with an “agitated” relative (11 [7–16]% vs. 23 [15–30]% of the time spent for resuscitation, *p* < 0.01). The teams confronted with an “agitated” relative showed more unsafe defibrillations, higher ventilation rates, and a delay in starting CPR (all *p* < 0.05 vs. control). The presence of a relative increased frustration, effort, and perceived temporal demands (all <0.05 compared to control); in addition, an “agitated” relative increased mental demands and total task load (both *p* < 0.05 compared to “withdrawn” and control group). The type of leadership condition did not show any effects. Conclusions: Interaction with a relative accounted for up to 25% of resuscitation time. Whereas the presence of a relative *per se* increased the task load in different domains, only the presence of an “agitated” relative had a marginal detrimental effect on CPR quality (GERMAN study registers number DRKS00024761).

## 1. Introduction

Discussing the presence of a relative during cardiopulmonary resuscitation (CPR) dates back to the 1980s and has been a controversial issue since [1,2,3,4]. Current guidelines recommend that resuscitation teams should offer family members of cardiac arrest (CA) patients the opportunity to be present during the resuscitation attempt in cases where this opportunity can be provided safely, and a member of the team can be allocated to provide support to the patient’s family [5,6,7]. Data on the effects of the different behaviour patterns of relatives (e.g., from withdrawn to agitated) on the quality of CPR and on the psychological impact on rescuers are sparse [8], and this trial aims at closing this gap.

Resuscitation per se is a stressful task [9]. Studies addressing the additional stress inflicted by family presence showed conflicting results covering an extent of no perceived additional strain to a sensed significant impairment of own activity due to family presence [1,4,10,11,12,13]. Regarding the quality of CPR, real-life studies and simulations have repeatedly shown high variability in CPR provision and sub-optimal compliance with guideline algorithms [14,15,16,17,18], but until now, however, no data reporting negative effects on patient outcomes have been published from institutions allowing relatives to be present during resuscitation [19,20]. Moreover, the presence of a “normally” behaving “unobtrusive” relative during simulated CA increased task load and frustration but did not impair the quality of CPR [13]. However, not all relatives can be expected to behave “normally” during the CPR of a loved one. Overtly aggressive or disruptive relatives or relatives who are withdrawn may well have a different impact on rescuers’ stress levels and the quality of CPR. It is mostly unknown if leadership (if any) may alleviate any impact of added strain caused by the presence of a relative.

Analysing the effect of a family member’s presence on the quality of resuscitation in a randomised controlled trial and under controlled conditions is quite challenging in reality, as the causes of cardiac arrest may vary from case to case. Simulations allow team performance to be studied in a realistic and standardised way [21], both overall and in individual subtasks, and performance metrics in simulation-based trials show a high level of consistency with resuscitation results. The object of our randomised, controlled, and prospective study was, therefore, to analyse the impact of family attendance with a withdrawn or agitated relative on the rescuer’s felt task burden as well as to assess leadership designation in this setting on the quality of CPR.

## 2. Materials and Methods

### 2.1. Participants

The Working Group on Intensive Care Medicine, Arnsberg, Germany (http://www.aim-arnsberg.de, assessed on 18 April 2022), arranges postgraduate educational, medical courses for physicians from Germany and German-speaking countries, mainly residents in the 2nd to 3rd year in internal medicine, Anaesthesiology, or surgery, who work in emergency and intensive care medicine. Attendees were offered participation in optional simulator-based training and were notified that the simulations would be video-recorded for scientific purposes. Physicians not willing to participate were offered identical workshops, but these were not filmed. The study was conducted according to the rules of the Declaration of Helsinki and authorised by the Ethics Committee of the Medical Association of Westphalia-Lippe (amendment to 2016-558-f-N, dated 6 September 2017), which waived the requirement to provide informed consent. The trial was registered with the German Clinical Trials Registry (www.drks.de, assessed on 18 April 2022; DRKS00024761) and is reported here according to the extensions of the CONSORT statements of the Reporting Guidelines for Health Care Simulation Research [22].

### 2.2. Study Design

This was a prospective, randomised, single-blind trial. The randomisations were carried out using computer-generated numbers. The participants were randomly assigned to teams of three to five physicians. The teams were then randomly allocated to perform CPR under three conditions: (1) no relative present, (2) “withdrawn” relative present, and (3) agitated relative present. A detailed description of the role can be found in the Appendix. In addition, the teams were randomly assigned so that either no leader was designated (no intervention), the team itself designated a leader (the rescue team received the mission of designating leadership before the scenario began), or the tutor designated a leader (leadership was attributed to a haphazardly selected group member by the tutor before the start of the scenario). The nominated leaders wore a coloured waistcoat and could thus be clearly recognised from the recordings. Aside from the attendance of a relative and allocated leadership, the requirements and conditions were the same for the teams.

### 2.3. Simulator and Scenario

The Ambu Man Wireless manikin (Ambu GmbH, Bad Nauheim, Germany) was employed. Attendants underwent a standardised implementation of the workshop, the manikin and the available CPR equipment. All members of the team were then briefed that their part during the subsequent setting was either that of an in- or out-of-hospital rescue team called to an unwitnessed cardiac arrest. The patient found (manikin) was pulseless, apnoeic, and had a Glasgow Coma Score of 3 (unresponsive, comatose). Once attached, ventricular fibrillation could be monitored on the defibrillator. The investigation period began when the first participant entered the room and stopped after the third defibrillation. The resuscitation manikins were served by trained tutors who were briefed not to perform any interventions until the investigation period ended.

### 2.4. Family Member Presence

A total of four actors were instructed to display two types of the patient’s relatives following scripted roles (see Appendix A). The attributes included quiet crying, mourning, and quiet observation for the “withdrawn” relative and loud crying or mourning as well as walking around the room worried and upset for the “agitated” relative. In order to ensure consistent quality in the course of the study, the video recordings were repeatedly discussed with all of the actors in the presence of an auditor. The teams assigned to the “relatives” groups met the relative at the scene of the incident beside the patient. The relative reported that the patient had collapsed in his presence and was unresponsive.

### 2.5. NASA Task Load Index

Subsequent to concluding the simulation, attendants were invited to fill out the NASA Task load index (NASA-TLX). The NASA-TLX was designed to evaluate the workload throughout or after a task. Six areas were rated after being assessed on visual analogue scales (0 to 100): Mental, Physical, and Time Demand, Frustration, Effort, and Own Performance [23]. It has been comprehensively ratified, is easily used and is comprehensively applied in various fields such as driving, teamwork, flying, and medicine [24,25].

### 2.6. Data Analysis

The data analysis was conducted by A.O., T.S. and S.M. using video recordings that were taken during the simulations. The starting point for the timing of all of the events was the first contact with the patient by one of the attendants.

### 2.7. Statistical Analysis

The primary outcome measure was the percentage of hands-on time, specified as the time of de facto chest compressions (CC), depicted as a fraction (in percent) of the total time available for CC. A power analysis adapted from pilot experiment data showed that it was necessary to study around 100 teams per study arm to identify a 10% difference between the groups in the primary outcome measure with a significance level of 0.05 (two-sided tested) and a power of 80%. Therefore, we determined to stop the trial once a minimum of 100 videotapes of adequate quality were available for each study arm. Due to organisational causes, the number of accessible video files of satisfactory quality could not be determined until the completion of the respective training course.

The secondary outcome measures involved the level of interaction with the relative, NASA TLX data, and compliance with different aspects of national and international guidelines on CPR. Additionally, the impact of the appointed leader was evaluated as a secondary outcome measure.

The complete data were scored on an intention-to-treat basis. The data are presented as medians [IQR (inter-quartile range)] unless otherwise reported. Statistical analysis was conducted using SPSS (version 25). Non-parametric ANOVA (Kruskal–Wallis test) was used for analysing the numerical data, subsequently followed by Mann–Whitney-U tests for independent samples where necessary. To obtain the estimates for differences between the medians and their approximate confidence intervals, the Hodges–Lehmann estimation was applied. The effect of leadership and the interactive term leadership × relative on outcomes was assessed using SPSS’s general linear model procedure with leadership assignment and relative assignment (i.e., study group) as fixed factors. The chi-square test was used for testing categorical data. A *p* < 0.05 (two-tailed) was considered statistically significant.

## 3. Results

### 3.1. Participants

After allocation and follow-up, the data of 335 teams (113 control, 117 “agitated”, and 105 “withdrawn”) were analysed (CONSORT flow chart, Figure 1). The teams consisted of 3 [3–4] physicians with no significant difference (*p* = 0.16) between the study groups. Verbal interactions with the “withdrawn” relative took place during a total of 11 [7–16]% of the study time and with the “agitated” relative in 23 [15–30]% (Difference 11% 95% CI 8–13; *p* < 0.01). Leadership had no influence on the allocation of interaction time (*p* = 0.54).

### 3.2. Primary Outcomes

Hands-on time was 91% [87–93] in the control group, 92% [88–94] in the “withdrawn” relative group, and 92% [88–93] in the “agitated” relative group (*p* = 0.15; Figure 2). Assigned leadership positions (*p* = 0.65) had no effect on hands-on time, and there was no significant relationship between hands-on and absolute (*p* = 0.64) or percentage of time (*p* = 0.54) spent verbally interacting with the relative.

### 3.3. Secondary Outcomes

The secondary outcomes are presented in Table 1. There were no differences between the control group and the withdrawn relative group for any quality marker of CPR. By contrast, compared to the control group, we observed a later start of resuscitation, more unsafe defibrillations, and higher ventilation rates in the agitated relative group. Leadership assignments showed no significant effect on the quality of CPR.

The NASA-TLX findings are summarised in Table 2 and Figure 3. Regardless of the relatives’ behaviour, the presence of a family member was related to higher scores for the domains temporal demand, effort, and frustration. In addition, “agitated” relatives increased the overall perceived task load and mental demand. The effects of leadership are summarised in Table 3.

## 4. Discussion

The present study demonstrates that regardless of their behaviour, the presence of family members during CPR increased rescuers’ perceived task load in several domains. While withdrawn relatives had no impact on the quality of CPR, the presence of an agitated relative was associated with minor effects on the start of CPR and defibrillation patterns.

Although family member presence during CPR is still a discussed and controversial issue [1,2,3,4], there is not yet sufficient evidence of the impact of the intervention on the outcome for the patient or family, and concerns about a performance effect exist among professionals and family members [7]. Data for relatives, who are belligerent or hinder CPR activities, are even more scarce [1,10,11]. As far as we know, this is the first study to quantify the burden of a rescue team during CPR dealing with a withdrawn or agitated relative.

### 4.1. Primary Outcome

By reemphasising the importance of high-quality chest compressions in the current guidelines, we selected hands-on time as the primary outcome variable [5]. We were unable to demonstrate a negative effect on hands-on time, which is in line with prior findings [13]. Interestingly, none of the papers on relatives’ presence during CPR quoted here commented on CPR quality itself [1,2,3,4,10]. Observational studies of facilities that allow relatives to be present during CPR found no obvious differences in patient outcomes after changing their protocols [20,26,27]. So far, there is only limited moderate to low-quality evidence suggesting that the presence of family members does not affect paediatric or adult CPR outcomes. The generalisation of these results beyond the out-of-hospital and emergency department setting is restricted owing to the lack of studies in alternative areas of health care [28]. A follow-up review appealing for further integration of relatives again describes no effects on CPR quality [29]. Although the controversial discussion regarding the presence of relatives during CPR may not be conclusively resolved, our present results show that at least CPR quality is not hampered and thus not a valid argument against their presence.

### 4.2. Secondary Outcomes

Honarmand and Crowley recently described a set of ACLS guideline deviations and their impact on outcomes from in-hospital cardiac arrest [30,31]. Selecting these deviations as secondary outcomes for the present trial, we observed no significant effect due to the presence of a “withdrawn” relative. For groups dealing with “agitated” relatives, statistically significant but clinically small effects were found for more unsafe defibrillations and higher ventilation rates as well as delays in starting CPR. These results are consistent with two previous trials where the attendance of a relative led to deferred defibrillation and fewer defibrillation attempts [11,13]. The observable slight delay in the onset of CPR could be a very subtle signal of initial distraction. However, as these findings are of little, if at all, medical relevance, we deduce that the physical attendance of a relative has no relevant negative influence on resuscitation quality.

### 4.3. NASA TLX

Emergency department personnel were reporting aggravated stress due to the attendance of relatives throughout resuscitation, and six out of twenty interviewees claimed to be hindered in their work [1,4]. In a post-incident survey offering options to answer ‘true’, ‘false’, or ‘I don’t know’, no difference in stress scores was found in relation to the attendance of relatives in a large clinical study [10]. Confirming the results of a prior trial with “normally” behaving family presence [13], the present trial found that family presence is associated with increased perceived temporal and mental demands and frustration of the rescuers involved. This effect is even more pronounced if confronted with an “agitated” relative.

In contrast to previous studies [10], our participants were able to provide more specific answers using the Likert scale of the validated NASA TLX [24]. In addition, it is rather uncommon for medical professionals to describe themselves as “stressed” in a dichotomous way, as this admission could give the impression of being weak or not resilient [13].

### 4.4. Strengths and Limitations

The identical conditions for all teams at all times in a large sample, together with the detailed evaluation of a large number of CPR tasks and subtasks right from the beginning of the scenario, are strengths of our simulation study. In contrast, there are often mentioned limitations in the context of a simulation study, such as the lack of real patients and—in the present study—also real family members. Due to the study centre and location, teams in the trial presented here comprised only physicians, so the results cannot necessarily be transferred to other team compositions.

## 5. Conclusions

Regardless of their behaviour, relatives present during CPR of a next of kin increase the perceived task load of rescuers involved. The presence of withdrawn relatives has no effect on CPR quality, while the presence of agitated relatives results in minor deviations only. Thus, the CPR quality is not an argument against family presence per se. In the case of massive disruption by agitated family members, teams might consider removing them from the setting to increase patient safety.

## Figures and Tables

**Figure 1 jcm-11-03163-f001:**
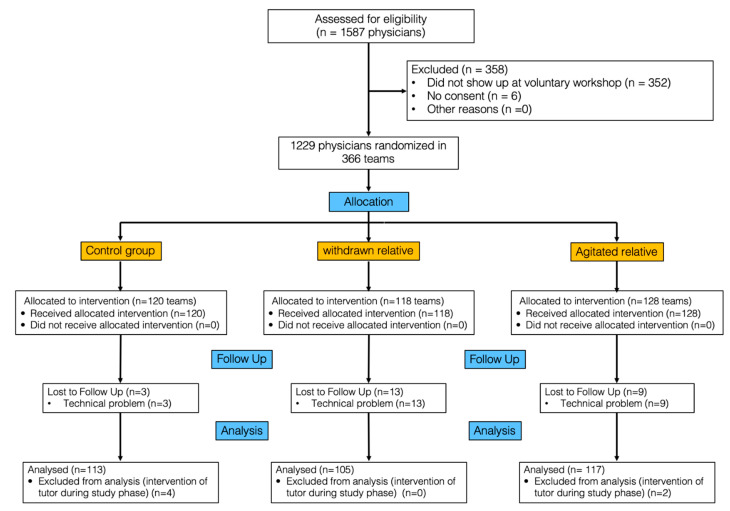
CONSORT flow chart.

**Figure 2 jcm-11-03163-f002:**
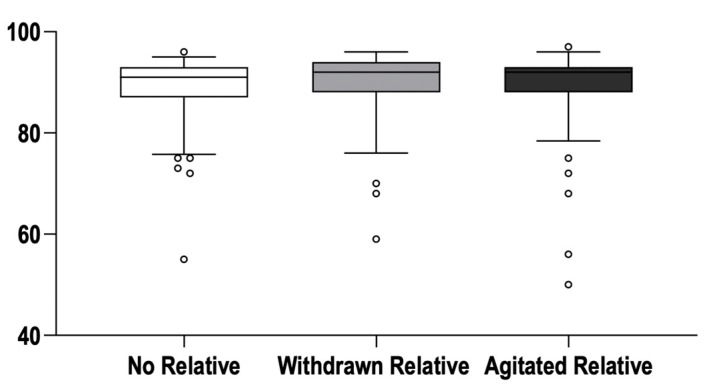
Primary outcome parameter (“Hands-on time”). Box and whisker plot of the percentage hands-on time. Boxes represent medians and interquartile range; whiskers delineate the 10th and 90th percentile, respectively; circles denote values outside the range from 10th to 90th percentile. There was no significant difference between the groups.

**Figure 3 jcm-11-03163-f003:**
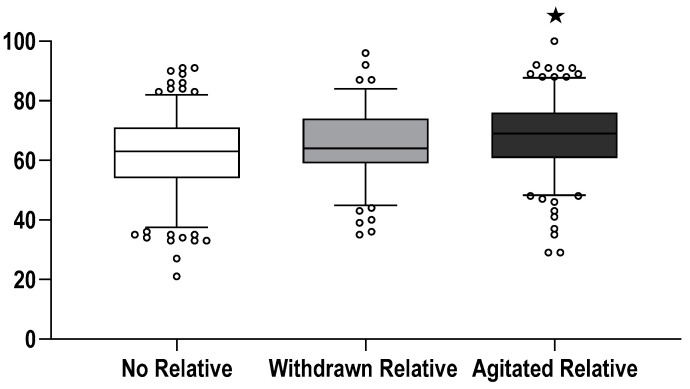
NASA TLX—weight average of components (according to [24]). Box and whisker plot of the perceived overall weighted task load. Boxes depict medians and interquartile range; whiskers outline the 10th and 90th percentile accordingly; circles denote values outside the range from 10th to 90th percentile. Ratings underlying the calculation of the task load are based on visual analogue scales ranging from 0 (minimum) to 100 (maximum). ★ In the agitated relative group, the task load was significantly higher than in the control group (*p* < 0.001) and the withdrawn relative group (*p* = 0.008). The difference between control and withdrawn relative group was of borderline significance (*p* = 0.058).

**Table 1 jcm-11-03163-t001:** Secondary outcome parameters.

	No Relative (Control) (*n* = 113)	Withdrawn Relative (*n* = 105)	Agitated Relative (*n* = 117)
**Chest compression**			
Time interval to CPR start (s)	14 [12–19]	17 [13–21]	18 [14–24] *
Start of CPR with CC (teams)	112/113	102/105	114/117
CC rates (strokes/min)	118 [112–125]	121 [112–126]	120 [111–127]
CC < 100/min (teams)	6/113	3/105	4/117
Change-overs per 2 min CPR (n)	1.4 [0.9–1.6]	1.5 [1.0–1.8]	1.3 [0.9–1.7]
**Defibrillation**			
Time to 1st defibrillation (s)	75 [55–102]	71 [52–103]	71 [53–106]
Shock with adequate (≥150 J) energy (teams)	113/113	105/105	117/117
VF not recognised ≥ once (teams)	4/113	2/105	4/117
Unsafe defibrillation ≥ once (teams)	30/113	31/105	54/117 *
**Airway Management**			
Advanced Airway Management (teams)	111/113	100/105	111/117
Time to Advanced Airway Management (s)	142 [95–194]	140 [111–205]	150 [105–214]
Advance airway position confirmed by capnography (teams)	89/111	85/100	90/111
Ventilation rate (b/min)	12 [4–10]	13 [8–19]	15 [10–20] *
**Medication**			
Time to i.v. line insertion (s)	112 [77–146]	93 [66–132]	111 [70–163]
Epinephrine administered (teams)	80/113	68/105	71/111
Correct dose (1 mg) administered (teams)	80/80	68/68	71/71
2nd dose after 3–5 min (teams)	2/6	0/6	2/5
Amiodarone administered (teams)	79/113	75/105	79/117
Correct dose (300 mg) administered (teams)	79/79	75/75	79/79
Administered after epinephrine AND 3rd shock (teams)	41/79	41/75	43/117
False ACLS drug administered (teams)	0/113	0/105	0/117

CC = chest compressions; ACLS = advanced cardiac life support; Data are medians [IQR] or proportions. Unsafe defibrillation was defined as not explicitly announcing the release of an electroshock or defibrillating while the patient was touched by a team member or a relative. Incorrect ACLS drugs are all drugs administered other than epinephrine and amiodarone. * = *p* < 0.05 vs. Control group.

**Table 2 jcm-11-03163-t002:** NASA Task load findings– single components.

	No Relative (Control) (*n* = 407)	Withdrawn Relative (*n* = 403)	Agitated Relative (*n* = 419)
Task Load	63 [54–71]	64 [59–73]	69 [61–73] *^,†^
Mental demand	70 [50–80]	70 [55–85]	75 [60–85] *^,†^
Physical demand	50 [30–70]	55 [40–75]	55 [35–70]
Temporal demand	70 [55–80]	75 [60–90] *	80 [65–90] *
Performance	55 [35–70]	55 [35–70]	55 [35–70]
Effort	65 [50–75]	70 [55–75] *	75 [60–85] *^,†^
Frustration	50 [30–70]	60 [45–75] *	65 [45–80] *^,†^

Data are medians [IQR]. Ratings are based on visual analogue scales ranging from 0 (minimum) to 100 (maximum). * = *p* < 0.05 vs. Control group; ^†^ = *p* < 0.05 vs. Withdrawn family member presence group.

**Table 3 jcm-11-03163-t003:** Effects of leadership assignment.

	Effect of Leadership Assignment	Effect of Relative Assignment	Interactive Term Leadership × Relative
**Chest compression**			
Hands-on time	0.65	0.55	0.09
Time interval to start of CPR	0.46	**0.015**	0.92
Start of CPR with massage	0.99	0.99	0.92
Chest compression rates	0.70	0.34	0.83
Compression rates < 100/min	0.92	0.83	0.89
Change-overs per 2 min CPR	0.43	0.08	0.67
**Defibrillation**			
Time to 1st defibrillation	0.38	0.52	**0.017**
VF not recognised ≥ once	0.99	0.99	0.98
Unsafe defibrillation ≥ once	0.31	**0.03**	0.62
**Airway Management**			
Advanced Airway Management	0.99	0.99	0.82
Time to Advanced Airway Management	0.45	0.59	0.31
Capnography to confirm airway position	0.87	0.67	0.20
Ventilation rate	0.72	0.06	0.16
**Medication**			
Time to i.v. line insertion	0.60	0.12	0.31
Time to epinephrine administration	0.46	0.12	0.60
Epinephrine administered	0.15	0.25	0.23
Amiodarone administered	0.46	0.79	0.36
Administered after epinephrine AND 3rd shock	0.25	0.82	0.94
**NASA Taskload**			
Task Load	0.98	**0.001**	0.31
Mental demand	0.35	**0.007**	0.29
Physical demand	0.60	0.08	0.74
Temporal demand	0.48	**0.001**	0.30
Performance	0.20	0.65	**0.003**
Effort	0.46	**0.001**	0.30
Frustration	0.09	**0.001**	0.09

Data are *p* values obtained from SPSS’s general linear model procedure with outcomes listed as dependent variables and leadership assignment and relative assignment (i.e., study group) as fixed factors. Unsafe defibrillation was defined as not explicitly announcing the release of an electroshock or defibrillating while the patient was touched by a team member or a relative. The significant interactive term for “time to 1st defibrillation” relates to swifter defibrillation in teams led by a tutor-designated leader in the agitated relative group; the significant interactive term for “performance” relates to higher ratings in teams without a designated leader in the agitated relative group. Bold: highlight the statistical significance.

## Data Availability

Not applicable.

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
