# Peer review of "The Impact of Withdrawn vs. Agitated Relatives during Resuscitation on Team Workload: A Single-Center Randomised Simulation-Based Study"

_jcm, 2022, doi:10.3390/jcm11113163_

Round 1
Reviewer 1 Report
Table 1. contains odd data: Withdrawn relative and No relaitve group Started CPR with massage 107/105 and 115/113 cases respectively. Please check.
Please provide exact definition on the drugs that of "Correct dose adminitered"
Fig 3.: please indicate the definition of the 0-100 scale of the box plot and indicate the significant difference among the groups on the figure directly.
The conclusion says: "the quality of CPR is not argument against family presence". Should the Authors refrase since the abstract and the data shown in the tables and figures indicate in connection with an “agitated” relative more unsafe defibrillations, higher ventilation rates and a delay in starting CPR was proven.
Author Response
Reply to Reviewer 1
We thank the reviewer for his thoughtful and constructive remarks. All suggestions and criticisms were incorporated in the revised version of our manuscript
Table 1. contains odd data: Withdrawn relative and No relaitve group Started CPR with massage 107/105 and 115/113 cases respectively. Please check.
Thank you for spotting this. We corrected these errors in the revised version
Please provide exact definition on the drugs that of "Correct dose adminitered"
The correct doses according to current guidelines of CPR for epinephrine (1mg) and amiodarone (300mg) are provided in table 1 as suggested
Fig 3.: please indicate the definition of the 0-100 scale of the box plot and indicate the significant difference among the groups on the figure directly.
Done as suggested
The conclusion says: "the quality of CPR is not argument against family presence". Should the Authors refrase since the abstract and the data shown in the tables and figures indicate in connection with an “agitated” relative more unsafe defibrillations, higher ventilation rates and a delay in starting CPR was proven.
Following the reviewer's suggestion, we have amended and supplemented as follows: “Thus, the quality of CPR is not argument against family presence per se. In the case of massive disruption by agitated family members, teams might consider removing them from the setting to increase patient safety.”

Reviewer 2 Report
Thank you for the great work. However I have few concerns regarding the manuscript
1. Most importantly , there are not sufficient information regarding the participants. Since this is a simulation study, the characteristics of participant and simulation setting, scenario is very important for interpretation. The author stated that the teams were consist of 3~5 physicians per a team, However since the outcomes were result of "TEAM" resuscitation rather than individual CPR and also task load that each individual felt, the actual number of physician in each team and mean number of physicians per a team according to the study group will be a very important confounding factor. However the authors did not present or adjust the factor during analysis. Please add the analysis of number of physician per team.
2.
Please add more explanation to agitated , withdrawn family member (although the author stated it was included in the appendix)
3.line 158 : "Leadership assignments (P = 0.64) had no effect on hands-on time and there was no significant correlation between hands-on time and the absolute time (P = 0.64) or percentage of time (P = 0.54) of verbal interaction with the family member." -> what kind of analysis did you use? could you present with additional table ?
4. Why did you set 2 intervention (type of family, type of leardership designation) in your study design? is there another analysis performed according to the leadership designation in a seperate publication? If the study design meant to have 2 interventions in a single simulation trial from the first place, the authors should decribe the results according to leardership designation in a proper table rather than just stating that there was no significant association.
5. the authors should describe more about their secondary outcome measures , including line 171 : what is the definition of "unsafe defibrillation"
6. Please add the type of statistic used in your table (such as median (IQR) , and the unit for each row (such as % , seconds .. )
Author Response
Reply to Reviewer 2
We thank the reviewer for his thoughtful and constructive remarks. All suggestions and criticisms were incorporated in the revised version of our manuscript
- Most importantly, there are not sufficient information regarding the participants. Since this is a simulation study, the characteristics of participant and simulation setting, scenario is very important for interpretation. The author stated that the teams were consist of 3~5 physicians per a team, however since the outcomes were result of "TEAM" resuscitation rather than individual CPR and also task load that each individual felt, the actual number of physicians in each team and mean number of physicians per a team according to the study group will be a very important confounding factor. However, the authors did not present or adjust the factor during analysis. Please add the analysis of number of physicians per team.
We agree with you that this information is important. Accordingly, the following sentence was added to the revised version of our manuscript: Teams consisted of 3 [3-4] physicians with no significant difference (p = 0.16) between study groups.
Additional information for reviewer 2: mean values ± standard deviation for team members of the control group, the withdrawn relative group, and the agitated relative group respectively were 3.37 ± 0.05, 3.28 ± 0.05, and 3.40 ± 0.05 with a p value of 0.191 in parametric ANOVA. Since data are not normally distributed and all other data in the manuscript are provided as medians [IQR] we choose to provide data on tem members in the medians [IQR] format.
- Please add more explanation to agitated, withdrawn family member (although the author stated it was included in the appendix)
We apologize for any inconvenience in case the reviewer has not received the appendix. According his proposal, we rephrased as follows:
“2.4. Family presence
Four actors were trained to play the two types of the patient’s relatives according to scripted roles (see Appendix part). Attributes included quiet crying, mourning and quietly observation for the “withdrawn” relative and loud crying or mourning as well as walking around the room worried and upset for the “agitated” relative.”
3.line 158: "Leadership assignments (P = 0.64) had no effect on hands-on time and there was no significant correlation between hands-on time and the absolute time (P = 0.64) or percentage of time (P = 0.54) of verbal interaction with the family member." -> what kind of analysis did you use? could you present with additional table?
The following sentence was added to the statistics’ section of the revised manuscript as suggested: The effect of leadership and the interactive term leadership*relative on outcomes was assessed using SPSS’s general linear model procedure with leadership assignment and relative assignment (i.e. study group) as fixed factors.
In addition, we added a new table (table 3) displaying the P values as obtained from general linear modelling for the effects of leadership assignment, family assignment, and the interactive term leadership*relative.
Additional information for reviewer 2: There is no non-parametric 2-way ANOVA available that provides interactive terms. For the present research, the interactive terms is important as leadership, even if overall not significantly relevant, might have a selective effect in the case of an agitated relative. SPSS’s general linear model procedure used is quite robust for non-parametric data.
- Why did you set 2 intervention (type of family, type of leadership designation) in your study design? is there another analysis performed according to the leadership designation in a separate publication? If the study design meant to have 2 interventions in a single simulation trial from the first place, the authors should describe the results according to leadership designation in a proper table rather than just stating that there was no significant association.
The aim of the study was to assess the effects of two different types of relatives AND of leadership designation on the quality of CPR and rescuers’ stress levels. We tried to make this aim clearer in the introduction. We rephrased as follows: “It is mostly unknown if leadership (if any) may alleviate any impact of added strain caused by the presence of a relative.” as we as “The object of our randomized, controlled and prospective study was therefore to analyse the impact of family attendance with a withdrawn or agitated relative on the rescuer's felt task burden as well as to assess leadership designation in this setting on the quality of CPR.”
Leadership could have an overall effect on quality measures of CPR (i.e. under all conditions) and/or a selective affect depending on a specific context (e.g. leadership is especially effective under more stressful conditions like an agitated relative). Such selective effect would become apparent by a significant P value for the interactive term between leadership and the context (in our study the relative assignment)
As suggested, the effects of leadership are displayed in a new table (table 3, see our comments above)
- the authors should describe more about their secondary outcome measures, including line 171: what is the definition of "unsafe defibrillation"
As suggested, all secondary outcomes (unless not immediately obvious) are explained in the revised table (either in the listing of outcomes or in the footnote)
- Please add the type of statistic used in your table (such as median (IQR) , and the unit for each row (such as % , seconds .. )
Tables are revised as suggested

Reviewer 3 Report
The discussed topic seems important.
The purpose of this original research is clear and consistent with the journal's field. The research questions were justified. Introduction properly introduces the reader into the undertaken issues.
The description and application of the measurement methods are appropriate.
The analysis of the results was carried out in detail. However, the presented results values are often given with too much accuracy.
The discussion was conducted in an appropriate manner. The final conclusions are justified and correct.
I believe that the manuscript has been prepared very carefully and I do not make any major remarks to its content.
Author Response
Reviewer 3
The discussed topic seems important.
The purpose of this original research is clear and consistent with the journal's field. The research questions were justified. Introduction properly introduces the reader into the undertaken issues.
The description and application of the measurement methods are appropriate.
The analysis of the results was carried out in detail. However, the presented results values are often given with too much accuracy.
The discussion was conducted in an appropriate manner. The final conclusions are justified and correct.
I believe that the manuscript has been prepared very carefully and I do not make any major remarks to its content.
We thank this reviewer for his/her supporting remarks.
